# Side Effects Associated with Probiotic Use in Adult Patients with Inflammatory Bowel Disease: A Systematic Review and Meta-Analysis of Randomized Controlled Trials

**DOI:** 10.3390/nu11122913

**Published:** 2019-12-02

**Authors:** Maria Pina Dore, Stefano Bibbò, Gianni Fresi, Gabrio Bassotti, Giovanni Mario Pes

**Affiliations:** 1Department of Medical, Surgical and Experimental Sciences, University of Sassari, 07100 Sassari, Italy; mpdore@uniss.it (M.P.D.); s.bibbo@gmail.com (S.B.); fresi.gianni@gmail.com (G.F.); 2Baylor College of Medicine, Houston, TX 77030, USA; 3Gastroenterology and Hepatology Section, Department of Medicine, University of Perugia, 06123 Perugia, Italy; gabassot@tin.it

**Keywords:** Crohn disease, ulcerative colitis, inflammatory bowel disease, probiotics, prebiotics, synbiotics

## Abstract

Probiotics demonstrated to be effective in the treatment of inflammatory bowel disease (IBD). However, the safety profile of probiotics is insufficiently explored. In the present systematic review and meta-analysis, we examined the occurrence of side effects related to probiotic/synbiotic use in randomized controlled trials (RCTs) of IBD patients as compared with placebo. Eligible RCTs in adult patients with IBD were identified by accessing the Medline database via PubMed, EMBASE, CENTRAL and the Cochrane central register of controlled trials up to December 2018. Occurrence of side effects was retrieved and recorded. Data were pooled and the relative risks (RRs) with their 95% confidence intervals (CIs) were calculated. The low-moderate study heterogeneity, assessed by the I^2^ statistic, allowed to use of a fixed-effects modelling for meta-analysis. Nine RCTs among 2337, including 826 patients (442 treated with probiotics/symbiotic and 384 with placebo) were analyzed. Eight were double-blind RCTs, and six enrolled ulcerative colitis (UC) patients. Although the risk for the overall side effects (RR 1.35, 95%CI 0.93–1.94; *I*^2^ = 25%) and for gastrointestinal symptoms (RR 1.78, 95%CI 0.99–3.20; *I*^2^ = 20%) was higher in IBD patients taking probiotics than in those exposed to placebo, statistical significance was achieved only for abdominal pain (RR 2.59, 95%CI 1.28–5.22; *I*^2^ = 40%). In conclusion, despite the small number of RCTs and the variety of probiotic used and schedule across studies, these findings highlight the level of research effort still required to identify the most appropriate use of probiotics in IBD.

## 1. Introduction

Inflammatory bowel disease (IBD) includes two major disorders: Crohn’s disease (CD) and ulcerative colitis (UC). With respect to the pathogenesis of IBD, the scientific literature suggests a fundamental role of the microbiota residing in the intestinal lumen and an inappropriate immune response to microbial factors [1,2]. The intestinal microbiota is acquired at birth—primarily influenced by delivery mode, changing rapidly during the first year of life according to nutritional factors such as breast feeding or artificial nutrition [3,4]. It is unique for each individual, although environmental factors including diet, drugs—especially anesthetics and antibiotics—stress and diseases may cause significant fluctuations [5]. Modifications of the quality and quantity of microbial communities may in turn contribute to intestinal inflammation, promoting IBD in genetically predisposed individuals [6]. On the other hand, intestinal microbiota can be modulated using probiotics, with attenuation of intestinal mucosal inflammation [7,8]. For example, in patients with UC, *Escherichia coli* 1917 Nissle was as effective as 5-acetyl salicylic acid in preventing relapse, and *Lactobacillus* GG was even more effective than mesalamine in prolonging relapse-free time [9,10]. The probiotic combination of VSL#3 was able to induce and maintain remission and reduce UC activity [11,12]. Moreover, *Lactobacillus reuteri* ATCC 55,730 showed to reduce clinical and endoscopic disease activity in children with UC [13]. In contrast, based on randomized controlled trials (RCTs) the efficacy of probiotics in CD is limited [14], although the addition of a prebiotic to *Bifidobacterium longum* showed a modest benefit in a small study [15]. Moreover, Kazemi et al. in patients with IBD detected a large decrease in C reactive protein and TNF-alpha cytokine following a treatment with probiotics [16]. In a recent study, we observed a reduction in the occurrence of major adverse outcomes related to IBD—requirement of treatment with systemic steroids, hospitalization, and surgery—in patients exposed to probiotic supplementation for 25–74% of the disease duration [17]. Although the US Food and Drug Administration recognized as safe some probiotics when added to food [18], the World Health Organization and the Food and Agriculture Organization of the United Nations in 2002 [19], reported that, theoretically, probiotics may be responsible for: Systemic infections, deleterious metabolic activities, excessive immune stimulation in susceptible individuals, gene transfer and minor gastrointestinal symptoms [20]. Hempel et al. reviewed 622 probiotics intervention studies in human for adverse effects; among them, 387 studies reported the presence or absence of specific adverse events including fungemia and bacteremia potentially associated with probiotic exposure [21]. Overall, in RCTs, the relative risk (RR) for gastrointestinal infections or other adverse events, was not significantly increased (RR 1.06; 95%CI: 0.97–1.16; *p* = 0.201) in patients exposed to probiotic compared with controls [21]. The authors conclude that “[…] despite the substantial number of publications, the current literature is not well equipped to answer questions on the safety of probiotic interventions with confidence”. For example, several studies and meta-analyses have assessed the efficacy of probiotics in IBDs. However, although data regarding side effects caused by probiotic have been reported by some authors [14], those related to meta-analyses are absent.

In the present systematic review and meta-analysis of placebo-controlled RTs, we evaluated the occurrence of side effects related to probiotic exposure in patients with IBD.

## 2. Materials and Methods

The protocol of this systematic review has been registered on the PROSPERO website [22] (PROSPERO registration number = CRD42019130389). We performed a systematic review and meta-analysis in accordance with the guidelines of the Preferred Reporting Items for Systematic Reviews and Meta-Analyses (PRISMA) [23]. Patients with IBD (Population) treated with probiotics (Intervention) or placebo (Comparator) in RCTs were analyzed, in order to evaluate the number of side effects (Outcome) in patients exposed to probiotics (PICO questions).

### 2.1. Search Strategy and Study Selection

A comprehensive search was conducted in the medical literature to identify all RCTs in adult patients with IBD by accessing the Medline database via PubMed, EMBASE, CENTRAL, and the Cochrane central register of placebo-controlled RTs up to December 2018. Studies were identified using the single terms “inflammatory bowel disease”, “Crohn disease”, “ulcerative colitis”, “pouchitis”, “colitis”, “ileitis”, “regional enteritis”, “probiotic”, “prebiotic”, “synbiotic”, “placebo”, “side effects”, and “adverse events” or a combination thereof using the boolean operators “AND” and “OR”.

The search was restricted to trials published in English as full peer-reviewed manuscripts. Systematic reviews on IBD and probiotics published before December 2018 were also manually screened for references of interest. The process was independently conducted by two reviewers (S.B. and G.F.), and identified studies were evaluated for analysis suitability. All RCTs considered eligible for the purpose of the present study were retrieved for further analysis. Eligible criteria defined a priori were: Peer-reviewed articles published in English, participants aged >16 years, randomized placebo-controlled trials, diagnosis of IBD based on endoscopy, histology or radiology, laboratory tests and clinician’s judgment, compared probiotics/synbiotics with placebo and dichotomous assessment of side effects occurrence. In addition, the references in each RCT were screened to double check for any missing study. Side effects were defined as symptoms and signs (including abnormal laboratory tests) judged by the authors as not directly related to IBD and for which a causal relationship with the study intervention was not ruled out. Risk of bias (RoB) was assessed according to the recommended categories in the Cochrane Handbook for Systematic Reviews of Interventions using the Risk of Bias 2.0 tool which assesses the effect of assignment to intervention [24].

### 2.2. Outcome Assessment

The assessed outcome was the number of patients with IBD exposed to probiotics who experienced the occurrence of any side effect, as compared with placebo.

### 2.3. Data Extraction

Information was extracted from each trial, independently and blinded, by S.B. and G.F., and potential discrepancies were discussed by all authors and resolved through consensus. Data included: First author’s name, year of publication, country of origin, study design, aim of the study, sample size, mean age and gender of trial participants, probiotic strain with or without prebiotic, dosage in colony forming unit (CFU), and schedule of probiotics. In addition, clinical data regarding IBD type, disease activity, and recorded side effects were collected. 

### 2.4. Data Synthesis and Statistical Analysis 

The meta-analysis was performed using Review Manager (RevMan) 5.3 software (Cochrane Collaboration) [25]. Side effects were expressed as a dichotomous variable (absence/presence) and compared between probiotic/synbiotic and placebo arms. Reported symptoms were categorized into subgroups, namely (i) gastrointestinal symptoms (nausea, abdominal pain, flatulence, bloating, unpleasant taste in the mouth, changes in stool, incontinence); (ii) abdominal pain; and (iii) others symptoms (cough, flu-like or coryza-like syndrome, arthralgia, fatigue, headache, stress, and depression among others).

The pooled effect of probiotics as predictor of side effects was calculated as RR with 95% confidence intervals (CIs). Heterogeneity was tested using the I^2^ statistic, a quantitative measure of inconsistency across studies. According to the Cochrane Handbook [26], moderate or no heterogeneity was considered when I^2^ was <50%, whereas substantial heterogeneity for I^2^ ≥ 50%. A *P* < 0.05 defined a significant degree of heterogeneity. In the absence of substantial heterogeneity (I^2^ < 50%), a fixed-effects model was used.

## 3. Results

### 3.1. Study Selection and Characteristics

The search strategy generated a total of 2337 articles, among which 2302 were evaluated as not pertinent to the study purpose (Figure 1). Of the remaining 35 studies, 14 articles published in the English language between 2002 and 2018 reporting side effects were considered relevant by both reviewers. Table 1 lists the characteristics of the selected studies. Five RCTs were further excluded because side effects occurrence was the same or absent in both arms, making RR not estimable [27,28,29,30,31]. Nine remaining studies [12,15,32,33,34,35,36,37,38], with a total of 826 patients, were pooled into the meta-analysis. Figure 2 shows the RoB assessment for each study according to specific domains. Risk of bias was low for all studies regarding the first and second domain as a result of blinding of participants and researchers. With the exception of three [12,35,37], in the remaining studies the side effects were not classified and/or graded, making the domain of outcome measurement at high risk/some concerns of bias.

Among these studies, there were 442 patients allocated to the probiotic/synbiotic arm and 384 to the placebo arm. The mean age of patients was 41.53 ± 5.81 years old in the probiotics group and 41.12 ± 7.13 years old in the control group. The proportion of women was similar in both arms (43.8% versus 43.5%, respectively). The most used probiotic was a mixture of *L. paracasei*, *L. plantarum*, *L. acidophilus*, and *L. delbrueckii* subsp bulgaricus, plus *B. longum*, *B. breve*, and *B. infantis*, plus one strain of *S. thermophilus* known as VSL#3. *Bifidobacterium longum*, or *B. breve* or *L. acidophilus*, or *L. johnsonii* or *E. coli* Nissle 1917, L. GG, B. bifidum and the Synergy synbiotic were also used (Table 1). All studies, except one, were double-blinded. Six studies included patients with UC and the remaining three included CD patients (Table 1).

### 3.2. Total Side Effects 

Total side effects included the three categories of (i) gastrointestinal symptoms; (ii) abdominal pain; and (iii) other symptoms. The pooled results showed an increased risk, albeit not significant, of total side effects in patients treated with probiotics/synbiotic as compared with those treated with placebo (72/442, 16.3% versus 32/384, 8.3% (RR 1.35, 95%CI 0.93–1.94), with low heterogeneity across studies (*I*^2^ = 25%; *p* = 0.22) (Figure 3).

A further analysis was restricted to double-blind RCTs to avoid the potential influence of experimenters on study participants. From the eight double-blind RCTs selected, there was a total of 805 patients involved (431 treated and 374 controls). Total side effects were reported by 71 patients treated with probiotics and 32 patients treated with placebo (Figure 3). Pooled results still showed an increased risk of side effects in treated patients (RR 1.33, 95%CI 0.92–1.91) with a moderate heterogeneity across studies (*I*^2^ = 32%; *p* = 0.17).

### 3.3. Side Effects According to IBD Type 

Six studies including 648 UC patients were pooled to evaluate side effects following probiotics administration. A significant increase in RR was observed in patients exposed to probiotics, with a moderate heterogeneity (38%, *p* = 0.16) across studies. The absolute risk (AR) was calculated and was equal to 0.176. 

There were three eligible trials including patients with CD showing no difference of developing side effects in patients exposed to probiotics/symbiotic as compared with those exposed to placebo (RR 1.07, 95%CI 0.52–2.19) (Figure 3).

### 3.4. Gastrointestinal Symptoms 

There were seven RCTs with a total of 735 patients reporting symptoms related to the digestive system (Figure 3). Overall, patients exposed to probiotics had an increased risk of complaining of gastrointestinal symptoms (RR 1.78, 95%CI 0.99–3.20). The calculated *I*^2^ (20%, *p* = 0.28) was not indicative of significant heterogeneity. More interestingly, when the analysis was further narrowed to the abdominal pain alone, the risk associated with probiotic use was even greater (RR 2.59, 95%CI 1.28–5.22), with a related heterogeneity of 40% and an AR of 0.085.

### 3.5. Fever and Respiratory Side Effects

Two RCTs in which probiotics were administered, reported fever and/or respiratory side effects in three patients, suggesting the absence of relationship between probiotic exposure and these side effects (Figure 3).

### 3.6. Excluded Studies

Five studies were not included in the meta-analysis because the RR was not estimable. More specifically, Kato et al. [27], Furrie et al. [28], and Yoshimatsu et al. [31], reported that the intervention was well tolerated and no one patients complained side effects. In the study conducted by Wildt et al. gastrointestinal symptoms were equally reported in both treatment groups, and a relation between intervention and side effects could not be established [30]. Ishikawa et al. despite observing side effects such as diarrhea or abdominal pain in patients exposed to synbiotics, did not report numbers, making the results unusable for the analysis [29].

## 4. Discussion

The results of our meta-analysis of RCTs show that patients with IBD exposed to probiotics experience side effects more frequently than those exposed to placebo.

There exists a robust body of literature on probiotics in patients with IBD; however, the majority of available studies are clinical trials designed to evaluate the efficacy of probiotics and prebiotics versus placebo, or other conventional therapies, for the achievement or maintenance of remission in patients with CD or UC [39]. In spite of that, the safety profile of probiotics and prebiotics in patients with IBD remains less explored, probably because they have been consumed as food for hundreds of years, especially in dairy products such as yogurt (the most famous are the Sardinian one: gioddu, the Caucasian: chefir; the Russian: kumis and the Egyptian: leben and for this reason perceived as safe. Described since ancient time in the Bible as a precious food, the acid milk regained popularity with the Nobel prize I.I. Mechnikov, who suggested its consumption against senility, up to nowadays with several probiotic strains, alone or combined, commercially available in hundreds of different products. Although the widespread use of probiotics, prebiotics and synbiotics, studies usually miss to report adequately adverse events related to their use, despite in some subgroups of patients their use was associated with severe side effects [20,21].

Our meta-analysis attempted, for the first time, to investigate the occurrence of side-effects in IBD patients undergoing treatment with probiotic/synbiotics and contrary to expectations, in a total of nine randomized placebo-controlled trials, we found that the risk of total side effects was higher in patients exposed to probiotics (RR 1.35) and the effect persisted (RR 1.33) when the analysis was restricted to double-blind RCTs. Interestingly, we observed a rising trend in the risk when the analysis focused on gastrointestinal symptoms (RR 1.78), or more specifically on abdominal pain (RR 2.59). These results are consistent with a previous meta-analysis addressing the benefit of probiotic in patients with UC and CD. Derwa et al. reported as a secondary endpoint of their meta-analysis a RR of 1.21 (95% CI = 0.64–2.27) to develop adverse events from six controlled RCTs in UC patients exposed to probiotics compared with placebo. The occurrence of adverse events reported by only one RCT did not reach a significant difference between CD patients treated with probiotics or placebo [14]. Similarly, in a systematic review and meta-analysis on the efficacy of probiotics, prebiotics, synbiotics and antibiotics in irritable bowel syndrome (IBS)among 36 trials with a total of 4183 patients Ford et al. found that patients treated with probiotics experienced adverse events more often as compared with those treated with placebo (19.4% versus 17.0%) although the RR was not significantly higher. Moreover, the authors detected significant heterogeneity among studies [40].

These findings are difficult to explain, since they are in contrast with the intrinsic meaning of probiotic “pro bios” in favor of life. For example, in a recent study conducted in a series of 200 IBD patients including both CD and UC, we observed a 93% reduction in the need for systemic steroids, hospitalization, and surgery related to the disease in CD patients, and a 100% reduction in UC patients taking probiotics for more than  75% of disease duration [17]. Similarly, in a study including 170 IBD patients exposed to probiotics use, we noticed a reduction in the occurrence of skin lesions [41]; however, this reduction was dependent on the amount of probiotics taken [17,41]. Probably, the duration of treatment with probiotics may influence outcomes. In the two previous studies probiotics were taken for years, whereas in the RCTs included in our meta-analysis they were taken for a maximum of one year [32,33]. In general, it is an increasingly accepted concept that probiotic benefits may be dose-dependent, only manifesting themselves upon achievement of a threshold dosage [42]. Probiotic amounts in currently used formulations show a wide variability, ranging from 10^7^ to 10^11^ CFU/g, which likely implies a high variability in the number of viable cells included in the products. It has been ascertained that the percentage of live cells capable of driving an effective change in fecal microbiota can vary from 1% to 92% [42]. We can assume that only formulations with a high bacterial load may exert a positive effect, while those with a low bacterial content have no effect or are even contrary to expectation. For example, daily doses of Lactobacilli equal to or greater than 10^10^ CFU induced a significant reduction in the duration of diarrhea in children. For lower doses of probiotics, an increase in the duration of diarrhea was paradoxically observed [43]. In addition to the duration and dose, the probiotic strain may play a central role in health and disease. For instance, in a double-blind placebo-controlled trial, Mangalat et al. observed a prominent proinflammatory triggering following the administration of *L. reuteri* in healthy adults as revealed by increased fecal calprotectin [44]. Moreover, some individuals taking probiotic may temporarily experience an increase in gas production and swelling, in addition to constipation, which in most cases disappears in a few weeks [45,46]. Several lactic bacteria produce bioactive substances such as histamine, tyramine, and phenylethylamine, which may induce headaches and other complaints [47]. These bioactive molecules are normally inactivated by mono-amino oxidases (MAO) in the intestinal wall and liver [48]. Minderhoud reported an alteration in neuroendocrine cells in IBD patients with IBS-like symptoms associated with lower MAO activity and an increase in biogenic amines [49]. A number of additional biogenic amines have also been isolated from bacterial strains commonly used in probiotic preparations [50]. These observations may provide a partial explanation for the occurrence of gastrointestinal side effects in patients exposed to probiotics including abdominal cramping, nausea, soft stools, flatulence, and taste disturbance, occurring in subjects receiving probiotics [20]. The gut microbiota resides almost completely in the colon and, to a lesser extent, in the small bowel. For this reason, it is plausible that probiotics affects especially colon microorganism communities. For the same reason treatment of IBD patients with probiotics is more effective in UC than in CD. In a review addressing the quantitative risk–benefit analysis of probiotic use in IBS and IBD, Bennet reported gastrointestinal symptoms as the most frequent side effects [51], although it is difficult to find a clear cut between gastrointestinal symptoms generated by the natural course of IBD and those generated by probiotic exposure. Another critical point is that in RCTs comparing probiotics with placebo in IBD patients, the conventional treatment was not similar in both arms, and in some series, the difference was statistically significant [15,33,35,37]. Lack of uniformity in the concomitant medications in each of these studies may be one limitation of this systematic review and meta-analysis. In addition, the majority of RCTs lacked standardized methods for the assessment of side effects (for instance, a validated questionnaire), or their graduation. Moreover, the strain, amount, schedule, and duration of probiotics/synbiotic treatment used in RCTs were extremely variable. Finally, but no less important, we were unable to relate a specific strain to side effect occurrence, due to the paucity of data.

## 5. Conclusions

In conclusion, at present, the small number of RCTs, the heterogeneity of the design and probiotic schedule across studies do not allow a satisfactory explanation of the apparent negative effect of probiotics in causing side effects in IBD patients. Future studies are required to identify the most appropriate species, strains or mixture thereof, and amounts of probiotics that are effective for IBD, while limiting potential side effects. To achieve this goal is fundamental that researchers measure and carefully report safety profile of probiotic/prebiotic/synbiotic in order to provide physicians guidelines to manage their patients with this treatment.

## Figures and Tables

**Figure 1 nutrients-11-02913-f001:**
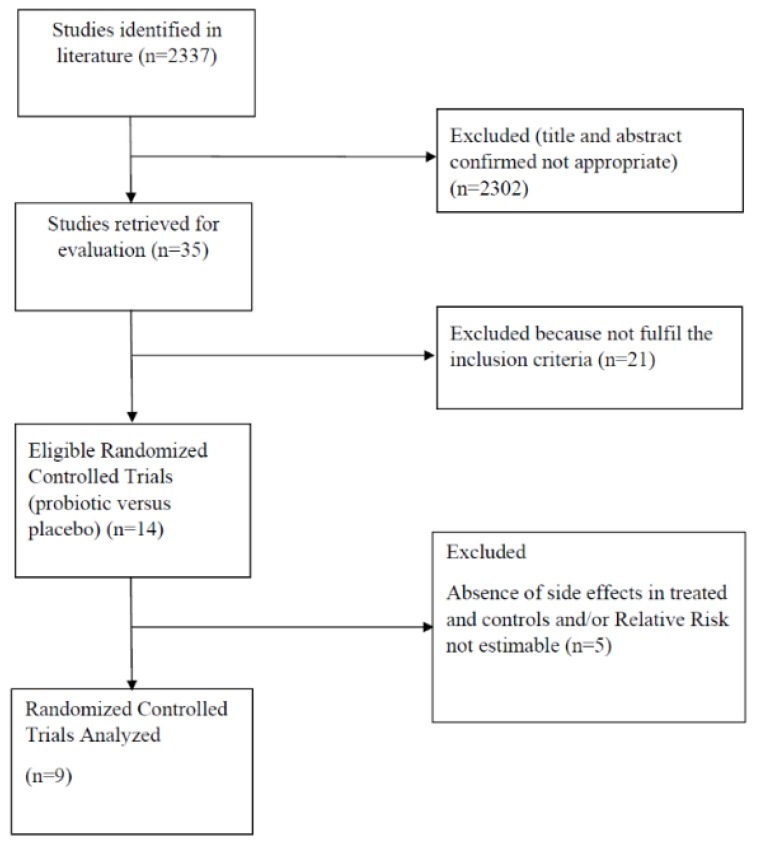
Flow diagram of assessment of studies identified in the systematic review and meta-analysis.

**Figure 2 nutrients-11-02913-f002:**
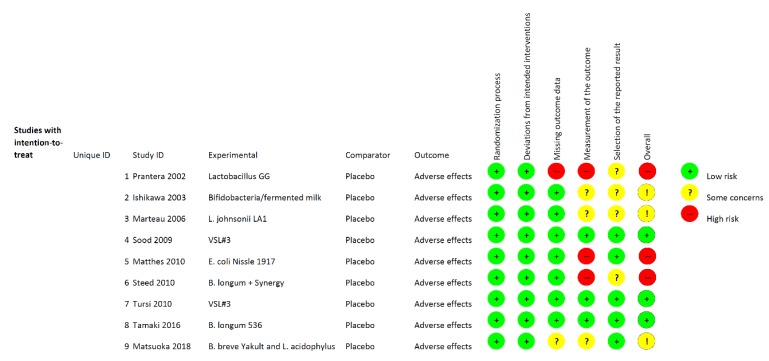
Risk of bias assessment of the randomized controlled trials included in the study.

**Figure 3 nutrients-11-02913-f003:**
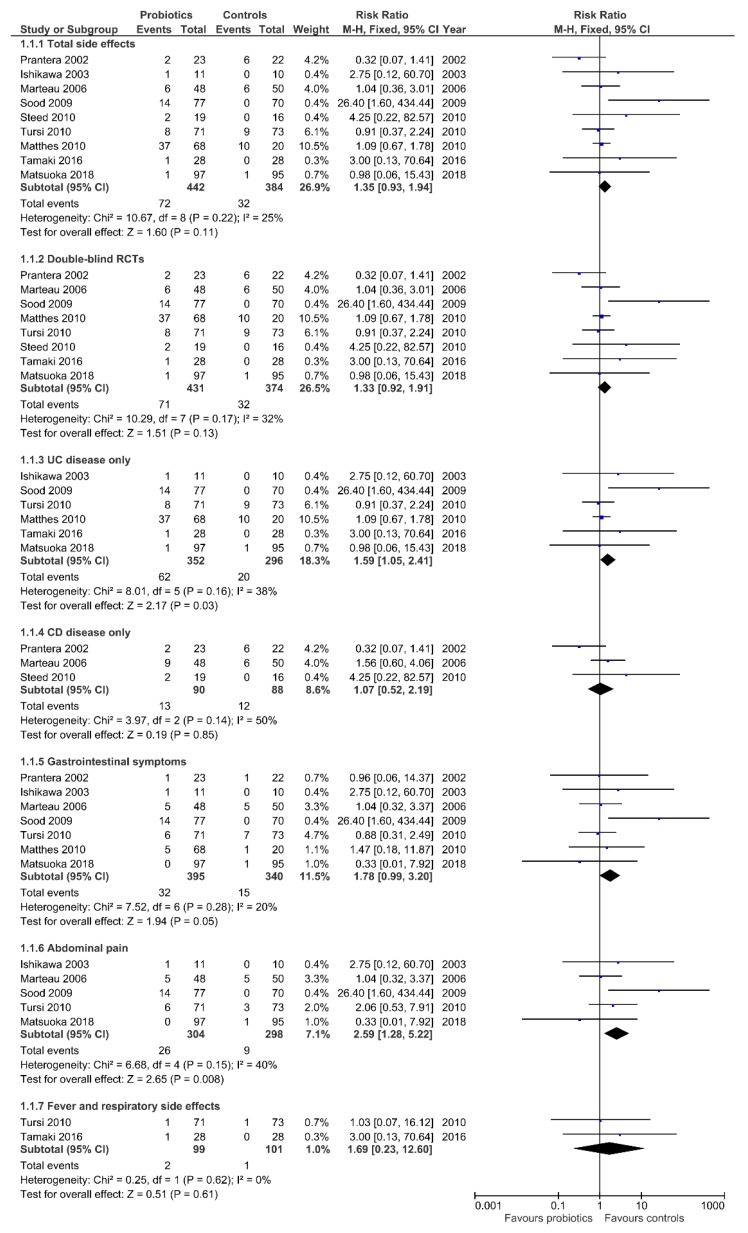
Forest plot of randomized controlled trials of probiotics/synbiotics vs. placebo in inflammatory bowel disease.

**Table 1 nutrients-11-02913-t001:** List of controlled clinical trials on adverse effects of probiotics versus placebo in inflammatory bowel disease.

Author, Year and Country	IBD Type	Intervention	No. of Patients	Dose and Duration	Side Effects	Study Design
Prantera et al., 2002 Italy [32]	CD	*Lactobacillus* GG	45	12 billion CFU/day for 52 weeks	Not described	DB, RCT
Ishikawa et al., 2003 Japan [33]	UC	Bifidobacteria-fermented milk (*B. breve,* and *B. bifidum*, and *L. acidophilus* YIT 0168 Yakult)	21	at least 10 billion per 100 mL bottle for 1 year	Coryza, Abdominal pain	RCT
Kato et al., 2004 Japan [27]	UC	*B. bifidum* + *B. breve* Yakult + *L. acidophillus*	20	10 billion per 100 mL/day for 12 weeks	No side effects in both arms	RCT
Furrie et al., 2005 UK [28]	UC	*B. longum* + Synergy	18	2 × 10^11^ CFU twice daily + 6 g fructo-oligosaccharide/inulin for 4 weeks.	No side effects in both arms	DB, RCT
Marteau et al., 2006 France [34]	CD	*L. johnsonii* LA1	98	2 × 10^9^ CFU 2packets/day for 6 months	Digestive disorders	DB, RCT
Sood et al., 2009 North India [35]	UC	VSL#3	147	3600 billion CFU/day for 12 weeks	Abdominal pain, Unpleasant taste	MC, DB, RCT
Matthes et al., 2010 Germany [36]	UC	*E. coli* Nissle 1917	90	10^8^ viable EcN/mL in 40, or 20 or 10 mL enemas/day for at least 2 weeks	Gastrointestinal disorders, Flatulence	Explorative, MC, DB, RCT
Steed et al., 2010 UK [15]	CD	*B. longum* + Synergy	35	2 × 10^11^ CFU twice daily + 6 g fructo-oligosaccharide/inulin for 6 months	Not better specified (2 patients unable to tolerate synbiotic)	DB, RCT
Tursi et al., 2010 Italy [12]	UC	VSL#3(*L. paracasei, L. plantarum, L. acidophilus, and L. delbrueckii subsp bulgaricus B. longum, B. breve, and B. infantis*, *Streptococcus thermophilus*)	144	3600 billion CFU/day for 8 weeks	Abdominal bloating and discomfort, Unpleasant taste, Dizziness, Flu-like symptoms	MC *, DB ^#^, RCT ^†^
Wildt et al., 2011 Denmark [30]	UC	*L. acidophilus* LA-5 + *B. animalis subsp. lactis* BB-12	32	1.5 × 10^11^ CFU/day for 52 weeks	Gastrointestinal symptoms equally reported in both arms	DB, RCT
Ishikawa et al., 2011 Japan [29]	UC	*B. breve* Yakult + GOS	41	10^9^ CFU/g three times daily + 5.5 g of galacto-oligosaccharide once a day for 1 year	diarrhea or abdominal pain. (without given number)	RCT
Yoshimatsu et al., 2015 Japan [31]	UC	1 tablet = *Streptococcus faecalis* T-110 (2 mg) + *Clostridium butyricum* TO-A (10 mg) + *Bacillus mesentericus* TO-A (10 mg)	60	9 tablets daily for 1 year	No side effects in both arms	DB, RCT
Tamaki et al., 2016 Japan [37]	UC	*B. longum* 536	56	2.3 × 10^11^ three times daily for 8 weeks	Dry cough	MC, DB, RCT
Matsuoka et al., 2018 Japan [38]	UC	*B. breve* Yakult (10 × 10^9^) + *L. acidophilus* (10^9^)	195	11 billion/day for 48 weeks	Body odor, Abdominal bloating, Stress	MC, DB, RCT

* multicenter trial; ^#^ double-blind trial; ^†^ randomized placebo-controlled trial; IBD: inflammatory bowel disease; CD: Crohn’s disease; UC: ulcerative colitis; CFU: colony forming unit.

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
