# Peer review of "Side Effects Associated with Probiotic Use in Adult Patients with Inflammatory Bowel Disease: A Systematic Review and Meta-Analysis of Randomized Controlled Trials"

_nutrients, 2019, doi:10.3390/nu11122913_

Round 1

Reviewer 1 Report

Manuscript ID: nutrients-631160

Title: Probiotic Use Increases Side Effects in Adult Patients with Inflammatory Bowel Disease: a Systematic Review and Meta-Analysis of Randomized Controlled Trials

Authors: Maria Pina Dore , Stefano Bibbò , Gianni Fresi , Gabrio Bassotti , Giovanni Mario Pes

This is a very interesting meta-analysis in which the authors examined the occurrence of side effects related to probiotic/synbiotic use in RCTs of patients with IBD as compared to placebo.  In general, the manuscript is very well written and addresses a clinically relevant topic. The authors clearly stated the objective in the introduction and performed a relatively comprehensive literature review.

The following concerns need to be addressed:

1-     I suggest that the authors reword the abstract to reflect the actual results; their CI is 0.93- 1.94 means, so there is no difference in the outcome overall. The only difference was when they focused their results to abdominal pain. Something similar to what they wrote in their conclusion, which was very balanced and objective, should be included in the abstract. I also suggest highlighting the difference in the results between UC and chron’s (see comment 2).

2-     The authors did a good job attempting to separate outcome due to disease (UC or Chron’s disease) and the type of intervention. However, this limitation needs to be more emphasized in all the sections of their manuscript. Their results show that UC patients on probiotics had more side effects compared to placebo, while chron’s disease did not. Therefore, I suggest that they highlight that difference in all their sections and try to explain why in the discussion.

Author Response

This is a very interesting meta-analysis in which the authors examined the occurrence of side effects related to probiotic/synbiotic use in RCTs of patients with IBD as compared to placebo.  In general, the manuscript is very well written and addresses a clinically relevant topic. The authors clearly stated the objective in the introduction and performed a relatively comprehensive literature review.

Reply: We thank the reviewer for his/her appreciation of our work.

The following concerns need to be addressed:

I suggest that the authors reword the abstract to reflect the actual results; their CI is 0.93- 1.94 means, so there is no difference in the outcome overall. The only difference was when they focused their results to abdominal pain. Something similar to what they wrote in their conclusion, which was very balanced and objective, should be included in the abstract. I also suggest highlighting the difference in the results between UC and Chron’s (see comment 2).

Reply: We thank the reviewer for his/her constructive observation. In the revised manuscript the abstract was changed accordingly (page 1, lines 19‒21 and 23‒29).

The authors did a good job attempting to separate outcome due to disease (UC or Chron’s disease) and the type of intervention. However, this limitation needs to be more emphasized in all the sections of their manuscript.

Reply: As reported in the discussion section (page 10, lines 266‒270) when citing the study of Bennett et al., the most frequent side effects are the gastrointestinal symptoms in patients exposed to probiotics, although it is difficult to distinguish those generated by the IBD and those generated by the use of probiotics. In our opinion to repeat this general statement in every section may generate redundancy.

Their results show that UC patients on probiotics had more side effects compared to placebo, while chron’s disease did not. Therefore, I suggest that they highlight that difference in all their sections and try to explain why in the discussion.

Reply: In the materials and methods section a definition of side effects was given: “Side effects were defined as symptoms and signs (including abnormal laboratory tests) judged by the authors as not directly related to IBD and for which a causal relationship with the study intervention was not ruled out” (page 3, lines 98‒100). An explanation as to why probiotics seem to have more side effects in Ulcerative Colitis than in Crohn’s disease is difficult and can be done only speculatively. In the revised manuscript we added the following sentence: “The gut microbiota resides almost completely in the colon and, to a lesser extent, in the small bowel. For this reason, it is plausible that probiotics affects especially colon microorganism communities. For the same reason treatment of IBD patients with probiotics is more effective in UC than in CD.” (page 10, lines 263‒266).

Reviewer 2 Report

In this systematic review and meta-analysis, Dore et al sought to quantify the risk of side effects (particularly focused on gastrointestinal symptoms) among IBD patients being treated with probiotics. The role of probiotics is gaining increasing attention. Although probiotics (e.g. VSL #3) have been showed to be helpful in the management of pouchitis among IBD patients,  data supporting a role of probiotics in induction and/or maintenance therapy in IBD are lacking. Studies evaluating the safety and side effect profiles of probiotics in IBD are equally important and clinically relevant. The scope of the work is appropriate for the journal Nutrients. However, I have major concerns about the conclusions and data-analysis that warrant attention. I have the following comments and suggestions that would improve the accuracy and quality of the manuscript:

-The author's title and conclusion are misleading and not supported by the data. The overall RR crosses 1 with insignificant p-value =0.11 for their primary outcome of total side effects. The risk of total side effects was only significant among ulcerative colitis (p=0.03) patients and for the outcome of abdominal pain in IBD patients (p=0.008) in subgroup analysis.

-Aside from low calculated heterogeneity, why did the authors choose a fixed-effects model rather than a random effects model for their meta-analysis? Under the fixed-effects model assumption, the only source of uncertainty is the within-study (sampling or estimation) error. This assumption clearly does not hold true for the included studies as there is significant variability between the different studies (type of probiotics used, duration of probiotics, IBD subgroup populations,etc) as well as variability and heterogeneity of clinical outcomes (GI symptoms plus other side effects). The authors should strongly consider employing a random-effects model instead.

-The authors should calculate absolute risks (number of side effects events/total number of patients) for their statistically significant outcomes. An increased risk of a rare event is not as clinically meaningful.

-The authors need to clarify what they meant by total side effects. Aside from GI symptoms and abdominal pain, did total side effects include other symptoms (coryza, stress, body odor, unpleasant taste, dizziness, etc)?

-Can authors provide data on more serious adverse events with probiotic use such as infections/sepsis and IBD flares? These outcomes are more clinically relevant and significant and tell more information about safety of probiotics as opposed to minor symptoms.

-What was the rationale to combine all GI symptoms into one major clinical outcome? Was abdominal pain included in this composite outcome?

-Can authors provide evidence that the patient's GI symptoms were not already present prior to initiation of probiotics? Can authors provide causation for their reported adverse events? How can authors be certain that the reported GI symptoms were not the result of an IBD flare as part of the natural history of the disease process versus actually caused by probiotics?

-In Table 1, authors should include relevant citations of included studies in relationship to their references section.

-How would the authors classify their primary outcome of side effects (major vs minor)? Are these major/clinically significant side effects to dissuade use of probiotics in IBD?

-How do the results of this study regarding risk of GI symptoms compare with pooled RR/OR of other meta-analyses reporting efficacy of probiotics in IBD? Authors should discuss risk-to-benefit ratio of probiotics in IBD.

-Given their findings (increased risk of total adverse events in ulcerative colitis, increased risk of the outcome of abdominal pain in IBD with probiotics), authors should provide a discussion with possible explanations/mechanisms. Why are ulcerative colitis patients differentially affected by probiotics? How do the authors think probiotics are causing increased abdominal pain? Is this from increased gas production from bacterial species? Are probiotics causing increased inflammation and IBD flares?

Author Response

In this systematic review and meta-analysis, Dore et al sought to quantify the risk of side effects (particularly focused on gastrointestinal symptoms) among IBD patients being treated with probiotics. The role of probiotics is gaining increasing attention. Although probiotics (e.g. VSL #3) have been showed to be helpful in the management of pouchitis among IBD patients,  data supporting a role of probiotics in induction and/or maintenance therapy in IBD are lacking. Studies evaluating the safety and side effect profiles of probiotics in IBD are equally important and clinically relevant. The scope of the work is appropriate for the journal Nutrients. However, I have major concerns about the conclusions and data-analysis that warrant attention. I have the following comments and suggestions that would improve the accuracy and quality of the manuscript:

-The author's title and conclusion are misleading and not supported by the data. The overall RR crosses 1 with insignificant p-value =0.11 for their primary outcome of total side effects. The risk of total side effects was only significant among ulcerative colitis (p=0.03) patients and for the outcome of abdominal pain in IBD patients (p=0.008) in subgroup analysis.

Reply: In the revised manuscript the title was changed as follows “Side Effects associated with Probiotic Use in Adult Patients with Inflammatory Bowel Disease: a Systematic Review and Meta-Analysis of Randomized Controlled Trials”.

-Aside from low calculated heterogeneity, why did the authors choose a fixed-effects model rather than a random effects model for their meta-analysis? Under the fixed-effects model assumption, the only source of uncertainty is the within-study (sampling or estimation) error. This assumption clearly does not hold true for the included studies as there is significant variability between the different studies (type of probiotics used, duration of probiotics, IBD subgroup populations,etc) as well as variability and heterogeneity of clinical outcomes (GI symptoms plus other side effects). The authors should strongly consider employing a random-effects model instead.

Reply: According to the Cochrane Handbook a substantial heterogeneity is present when I² is equal to or higher than 50% (Cochrane Handbook ver. 5.1 Section 9.5.2, page 278). Since in our analysis we selected only double blind RCT, while observational studies were excluded, between‒study heterogeneity was a priori low.

In our opinion there was no specific reason to adopt a random‒effect model since there was no obvious source of unexplained heterogeneity such as design diversity, different inclusion criteria or treatments given in different way. Hence, we are fairly confident that, under the assumption that all studies were sufficiently homogeneous, we were estimating the same intervention effect.

The more conservative random-effects approach, suggested by the reviewer, was avoided because we feel that the accurate selection procedure of the studies included guaranteed sufficient homogeneity. We observe incidentally that the fixed effects approach does not require that the study effect should actually be the same and we were not interested in the mean effect size for the whole “universe” of possible studies but only in the specific effect size of the studies included in our metanalysis (see Hunter and Schmidt, Int J Select Assess, 2000;8:275-292).

The decision to use the fixed effect model was dictated, in addition to the finding of a heterogeneity lower than 50%, also by the fact that the studies included in the analysis were mutually consistent, co-directional and methodologically homogeneous. The use of a random effect approach would have unnecessarily made more “fuzzy” the size effect estimates.

-The authors should calculate absolute risks (number of side effects events/total number of patients) for their statistically significant outcomes. An increased risk of a rare event is not as clinically meaningful.

Reply: In the revised manuscript the absolute risk for side effects was calculated in the two outcomes that resulted significant (page 6, lines 170‒171 and 181).

-The authors need to clarify what they meant by total side effects. Aside from GI symptoms and abdominal pain, did total side effects include other symptoms (coryza, stress, body odor, unpleasant taste, dizziness, etc)?

Reply: In the original manuscript this was clearly stated at page 3, lines 115‒120 where we stated that total side effects embrace gastrointestinal symptoms, abdominal pain and other symptoms including coryza, stress, body odor, unpleasant taste, dizziness. In addition, this was better specified at page 6, lines 156‒157.

-Can authors provide data on more serious adverse events with probiotic use such as infections/sepsis and IBD flares? These outcomes are more clinically relevant and significant and tell more information about safety of probiotics as opposed to minor symptoms.

Reply: All symptoms related to probiotic use and reported by the RCT authors have been mentioned in the original manuscript.

-What was the rationale to combine all GI symptoms into one major clinical outcome? Was abdominal pain included in this composite outcome?

Reply: Of course, abdominal pain should be definitely considered a GI symptom. The rationale of combining all GI symptoms in a single outcomes relies on the idea to provide gastroenterologists (who mainly manage IBD cases) with a comprehensive category of risk.

In the original manuscript this was specified at page 3, lines 115‒120.

-Can authors provide evidence that the patient's GI symptoms were not already present prior to initiation of probiotics? Can authors provide causation for their reported adverse events? How can authors be certain that the reported GI symptoms were not the result of an IBD flare as part of the natural history of the disease process versus actually caused by probiotics?

Reply: As explained earlier to the reviewer no. #1 in the original manuscript we specified that… “Side effects were defined as symptoms and signs (including abnormal laboratory tests) judged by the authors as not directly related to IBD and for which a causal relationship with the study intervention was not ruled out” (page 3, lines 98‒100) and in the discussion section (page 10, lines 266‒270) …………….the most frequent side effects are the gastrointestinal symptoms in patients exposed to probiotics, although it is difficult to distinguish those generated by the IBD and those generated by the use of probiotics.

Our meta-analysis was based on the data reported by the authors of the selected studies without casting doubt on their main findings.

-In Table 1, authors should include relevant citations of included studies in relationship to their references section.

Reply: Done.

-How would the authors classify their primary outcome of side effects (major vs minor)? Are these major/clinically significant side effects to dissuade use of probiotics in IBD?

Reply: Although Doron. et al (20) reported a description of “systemic infection deleterious metabolic activity eccessive in immune stimulation in susceptible individuals, and gene transfer that may pose a clinical contraindication for probiotic use in specific populations” (page 2, lines 59-61), this was not our case. All side effect analyzed in this meta-analysis have been considered as being “minor”. The “take home” message of our meta-analysis is the one summarized at page 10, lines 282-286.

-How do the results of this study regarding risk of GI symptoms compare with pooled RR/OR of other meta-analyses reporting efficacy of probiotics in IBD? Authors should discuss risk-to-benefit ratio of probiotics in IBD.

Reply: See above.

-Given their findings (increased risk of total adverse events in ulcerative colitis, increased risk of the outcome of abdominal pain in IBD with probiotics), authors should provide a discussion with possible explanations/mechanisms. Why are ulcerative colitis patients differentially affected by probiotics? How do the authors think probiotics are causing increased abdominal pain? Is this from increased gas production from bacterial species? Are probiotics causing increased inflammation and IBD flares?

Reply: In the manuscript we already discussed several potential mechanisms that were mentioned in the literature. Upon the base of our reflection a sentence was added in the discussion section of the revised manuscript “The gut microbiota resides almost completely in the colon and, to a lesser extent, in the small bowel. For this reason, it is plausible that probiotics affects especially colon microorganism communities. For the same reason treatment of IBD patients with probiotics is more effective in UC than in CD.” (page 10, 263-266).

Round 2

Reviewer 2 Report

The authors have adequately responded to my critiques and recommendations. The manuscript is appropriate for publication in Nutrients.